# Active control of flexible spacecraft in orbit based on partial differential equations

**Bo Zhang**[1]*, **Ming Wen**[2]

**1** Hunan Mechanical and Electrical Polytechnic, Changsha, Hunan, China, **2** Hunan Mingxiang Aviation Technology Co., Ltd., Changsha, Hunan, China

* bozhang1986edu@yeah.net

**Data availability statement:** All relevant data are within the paper and its Supporting information files.

## Abstract

Flexible spacecraft possess the ability to adapt to complex environments and use energy more efficiently, offering enhanced flexibility and stability in space missions, particularly in tasks with significant external disturbances such as deep space exploration and satellite attitude control. However, vibration suppression in flexible spacecraft remains a critical challenge. This study addresses the problem of vibration suppression in flexible spacecraft systems under external disturbances and input constraints. First, a partial differential equation (PDE) with boundary initial conditions is derived using Hamilton's principle, accurately describing the dynamic characteristics of the flexible structure. A backstepping controller based on the Nassbaum function and a disturbance observer is then designed to ensure system stability in the presence of input constraints and external disturbances. A Lyapunov function is constructed, and appropriate control parameters are selected to further guarantee system stability. Numerical simulations confirm the superiority of the proposed control method, with results showing an $80\%$ reduction in settling time and a $94\%$ decrease in peak overshoot compared to conventional PD control. The proposed scheme significantly enhances the performance and stability of flexible spacecraft systems, demonstrating its potential for improving spacecraft dynamics in challenging space environments.

## 1 Introduction

A flexible spacecraft refers to a spacecraft with a flexible or deformable structural design [1–3]. These spacecraft are typically made of lightweight, deformable materials that can adapt to changes in the external environment during operation, such as air flow, gravity, temperature variations, and vibrations during maneuvers. Vibration suppression for flexible spacecraft is necessary because their flexible structure is prone to unnecessary vibrations caused by external disturbances or operations. These vibrations can affect the spacecraft's stability and precise control, thereby impacting the execution of the mission. Vibration suppression helps reduce such interference, ensuring that the spacecraft operates more stably during the mission, enhancing its performance and reliability[4–6].

Comparing with the traditional hard spacecraft, the flexible spacecraft system has complex dynamic characteristics, so it is more difficult to design the controller (see [7,8]). The

**Funding:** The author(s) received no specific funding for this work.

flexible spacecraft is essentially an infinite dimensional distributed parameter system (DPS), and its state variables (space and time) are generally described by partial differential equations (PDEs). The previous research on the dynamic characteristics of flexible hose is based on ordinary differential equations (ODEs) (see [9]). Different from other traditional DO (see [10,11]), Takagi-Sugeno fuzzy method is used to describe the flexible model of flexible solar panels. The simulation also proves that this method has good stability and robustness. Awan, Zainab Shahid and Ali, Khurram et al. proposed a hybrid fault-tolerant control scheme for a 6-DOF robotic manipulator, integrating nonlinear observers, sensor redundancy, and adaptive backstepping-based actuator fault estimation with controller reconfiguration, validated by LabVIEW simulations for improved tracking under faults[12]. In [13], according to the dynamic characteristics of microsatellite, a new compound controller based on sliding mode control and finite time DO is proposed in the inner loop. In [14], a finite-time trajectory tracking controller is presented for a space manipulator, addressing model uncertainty, external disturbances, and actuator saturation. A radial basis function neural network (NN) is used to estimate model uncertainties, while an auxiliary system compensates for actuator saturation, and a NN-based adaptive terminal sliding mode controller ensures stable trajectory tracking, as confirmed by Lyapunov stability analysis and numerical simulations. In [15], a robust nonlinear hybrid control for a MIMO separately excited DC motor, combining adaptive backstepping and integral sliding mode control to handle uncertainties and disturbances, with simulation results confirming improved tracking and reduced settling time compared to conventional methods. Although ODEs are simple in model description and controller design, it can not accurately describe the physical dynamic characteristics of flexible systems, which easily leads to control overflow.

There are many research results on the description of system dynamic characteristics based on PDEs. In [16], this study presents two distributed control strategies for a team of flexible spacecraft to track the attitude of a virtual leader using a PDE observer. The first strategy uses direct state measurements from the hub and free tip of the beam, while the second incorporates a PDE observer to estimate the beam's states, with both approaches ensuring asymptotic attitude tracking without residual vibration. In [17], the flexible satellite dynamics is modeled as a combination of coupled ordinary differential equations (ODEs) and PDEs using Hamilton's principle. Then, a fault-tolerant control (FTC) strategy based on adaptive integral sliding mode is proposed to address problems such as inertial uncertainty and external disturbances. In [18], the use of basic PDE models and boundary control helps simplify the dynamic modeling of a small robot with flexible wings. Similarly, [19] derives nonlinear coupled PDEs for the dynamics of a satellite propellant tank using Hamilton's principle, and applies a PDE-based controller to suppress slosh instability. In [20], flexible solar panels are modeled as symmetric Euler-Bernoulli beams using PDEs, and a composite controller combining Nussbaum-type functions and backstepping is developed to suppress vibration under input constraints and external disturbances. Reference [21] also adopts a PDE-based infinite-dimensional modeling framework for flexible structures, similar to [17] and [20], and designs a boundary control-based FTC scheme.

While PDE-based modeling enables high-fidelity representation of distributed-parameter systems, especially in flexible structures and fluid-structure interactions, it inevitably increases the computational complexity. Compared to ODE-based lumped models, PDE models result in infinite-dimensional systems that require spatial discretization techniques (e.g., finite element, finite difference, or spectral methods) for numerical simulation and control implementation. Moreover, there are many literatures on control methods for suppressing flexible spacecraft in the presence of external disturbances, such as disturbance observer based control (DOBC) (see [22]), active disturbance rejection control (ADRC) [23,24] and so on.

In [25], the complex dynamic model of refueling hose is described based on PDEs. Then, a DO is designed to estimate the vibration of the hose. In [26], PDEs and boundary control method are used to suppress the vibration problem of nonlinear flexible manipulator in three-dimensional space. Besides, the output signal is controlled within the adjustable range by selecting the barrier Lyapunov function. In [27], according to Hamilton's principle and PDEs, the refueling hose model transformed into a DPS. Furthermore, a boundary control scheme is proposed based on the original PDEs to regulate the vibration of the flexible spacecraft and deal with the influence of control input constraints.

Although numerous studies have addressed the characteristics of flexible systems using PDEs (see [10,20,28]), the application of PDEs to describe the characteristics of flexible spacecraft is limited due to the complexity involved in controller design. In practical engineering, many physical characteristics cannot be adequately described using ordinary differential equations (ODEs). Flexible systems inherently possess infinite dimensions and are essentially distributed parameter systems (DPS). Consequently, the flexible hose model can be represented by both infinite-dimensional equations (which describe the flexible body using PDEs) and finite-dimensional equations (which describe the boundary conditions using ODEs). The use of PDEs to model flexible systems introduces challenges in controller design. Therefore, developing a novel control strategy to address the design challenges associated with flexible hose systems is of significant practical importance.

Inspired from the above mentioned documents, in this paper, a novel controller based on boundary control is proposed for flexible hose subject to the external disturbances and input constraints. The main contributions of this paper are as follows:

1. Different from describing model characteristics based on traditional ODEs (see [22,29]). Flexible spacecraft have infinite dimensions and are essentially DPS. ODEs can not accurately describe the system characteristics, and even cause control overflow problems. Thus, this paper describes the dynamic characteristics of flexible hose based on PDEs.
2. Considering the external disturbances and input constrainted in flexible hose system, a backstepping control scheme (see [20,30–33]) based on DO and boundary conditions is proposed for suppressing the elastic vibration $\eta(x, t)$ of the flexible spacecraft .
   i. The disturbance observer is constructed to estimate the disturbance $d(t)$ in the flexible hose system, which further enhances the anti-disturbance performance of the system.
   ii. For the convenience of controller design, DO and Nussbaum functions are constructed to deal with the problem of external disturbances and input constraints.
3. With the proposed boundary control strategy, the closed-loop system can be uniformly bounded by Lyapunov direct method, and the hose state of the system can be converged to a compact set by selecting appropriate parameters.

The structure of the paper is organized as follows: In Section 1, we provide an overview of the problem description and the analysis of the flexible hose model using PDEs. Section 2 introduces a novel backstepping control strategy that utilizes boundary conditions and DO to enhance control performance. In Section 3, we present a comparative study between the proposed control method and a traditional PD control approach through simulation examples, demonstrating the effectiveness and advantages of our proposed strategy. Finally, Section 4 offers concluding remarks and discusses potential directions for future work in this field.

## 2 Problem description and model analysis

### 2.1 Problem description

The flexible spacecraft system is mainly composed of tanker, receiver and flexible spacecraft, as shown in Fig 1. $X_g$–$Y_g$ represents an inertial reference frame. $X$–$Y$ represents the local coordinate system and moves horizontally. $x$–$y$ is the body fixed coordinate system, which is attached to the connection point between the flexible spacecraft and the receiver aircraft. In this paper, only the horizontal direction of flexible spacecraft is discussed. The control input $u(t)$ represents the top boundary actuator of the flexible spacecraft. $d(t)$ represents the external disturbances to the flexible hose during flexible spacecraft. $m$ is the mass of the actuator.

### 2.2 Dynamic analysis

Compared with the traditional HPAR, the flexible spacecraft system belongs to DPS in essence. According to Hamilton's principle (see [34–36]), the dynamic model of flexible hose can be obtained by variation as:

$$\int_{t_1}^{t_2} \left( \sigma \Sigma_k(t) - \sigma \Sigma_p(t) + \sigma \Sigma(t) \right) dt = 0 \tag{1}$$

where $\sigma(\cdot)$ represents the variation of $(\cdot)$. $t_1$ and $t_2$ are two time constants and satisfy $0 < t_1 < t_2$. $\Sigma_k(t)$, $\Sigma_p(t)$ and $\sigma \Sigma(t)$ represent the kinetic energy (KE), potential energy (PE) and virtual work of the system respectively. It is worth noting that $\sigma \Sigma(t)$ is the virtual work done by the non-conservative force, which includes the lateral load of the hose system, external disturbances and flexible damping structure.

The KE of the hose system can be written as

$$\Sigma_k(t) = \frac{1}{2}\rho \int_0^l \left( \frac{\partial \eta(x,t)}{\partial t} \right)^2 dx + \frac{1}{2} m \left( \frac{\partial \eta(x,t)}{\partial t} \right)^2 \bigg|_{x=l} \tag{2}$$

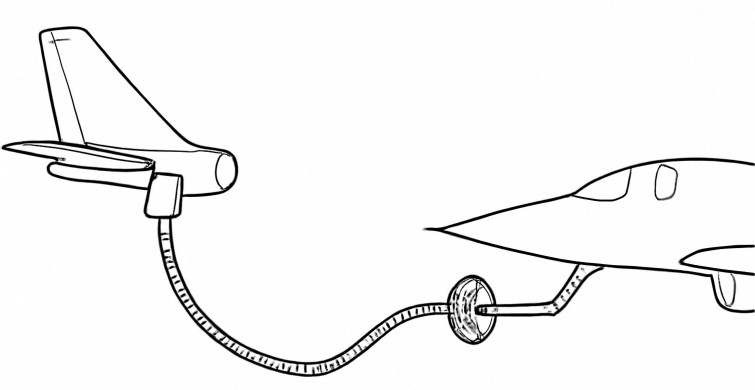

**Fig 1. The diagram of flexible spacecraft system.**

where $\eta\,(x,t)$ is the elastic deflection in time $t$ relative to the position $x$. $\rho$ represents the density of hose material. $l$ represents the length of the flexible spacecraft. The $\Sigma_p\,(t)$ due to the bending can be obtained from

$$\Sigma_p\,(t) = \frac{1}{2}EI\int_0^l\left(\frac{\partial^2\eta(x,t)}{\partial x^2}\right)^2 dx \tag{3}$$

where $E$ represents young's modulus, $I$ represents the inertia matrix of hose. For clarity, defining $\Theta = EI$.

The virtual work done by the flexible hose system is expressed as

$$\sigma\Sigma\,(t) = -\int_0^l\Gamma_1\frac{\partial\eta\,(x,t)}{\partial t}\sigma\partial\eta\,(x,t)\,dx \tag{4}$$

where $\Gamma_1$ represents elastic damping coefficient.

It can be written that the virtual work done by generating the hose lateral control force of $u(t)$ to suppress vibration can be expressed as

$$\sigma\Sigma_u\,(t) = \sigma u\,(t)\,\eta\,(x,t) \tag{5}$$

From Eqs (2) and (3), we obtain

$$\sigma\Sigma_k\,(t) = \rho\int_0^l\frac{\partial\eta\,(x,t)}{\partial t}\sigma\frac{\partial\eta\,(x,t)}{\partial t}dx + m\frac{\partial\eta\,(x,t)}{\partial t}\sigma\frac{\partial\eta\,(x,t)}{\partial t}\Big|_{x=l} \tag{6}$$

$$\sigma\Sigma_p\,(t) = \Theta\int_0^l\frac{\partial^2\eta\,(x,t)}{\partial x^2}\sigma\frac{\partial^2\eta\,(x,t)}{\partial x^2}dx \tag{7}$$

and we further obtain

$$\int_{t_1}^{t_2}\sigma\Sigma_k\,(t)dt = -\rho\int_{t_1}^{t_2}\int_0^l\frac{\partial^2\eta\,(x,t)}{\partial t^2}\sigma\eta\,(x,t)\,dxdt - m\int_{t_1}^{t_2}\frac{\partial^2\eta\,(x,t)}{\partial t^2}\sigma\eta\,(x,t)\Big|_{x=l}dt \tag{8}$$

$$\int_{t_1}^{t_2}\sigma\Sigma_p\,(t)dt = \Theta\int_{t_1}^{t_2}\int_0^l\frac{\partial^4\eta\,(x,t)}{\partial x^4}\sigma\eta\,(x,t)\,dxdt + \Theta\int_{t_1}^{t_2}\left[\frac{\partial^2\eta\,(x,t)}{\partial x^2}\sigma\frac{\eta\,(x,t)}{\partial x} - \frac{\partial^3\eta\,(x,t)}{\partial x^3}\sigma\eta\,(x,t)\right]\Big|_0^l dt \tag{9}$$

For $\forall_x\in[0,l]\,,t\in[0,\infty)$, by combining the derived equations with appropriate boundary conditions and applying Hamilton's principle (Eq 1), we can obtain the structural dynamics of the flexible spacecraft as follows

$$\rho\eta"\,(x,t) + \Theta\eta_{xxxx}\,(x,t) + \Gamma_1\eta\dot{\,}\,(x,t) = 0 \tag{10}$$

$$\eta_x\,(0,t) = \eta_{xx}\,(0,t) = \eta_{xxx}\,(0,t) = 0 \tag{11}$$

$$m\eta"\,(l,t) = \Theta\eta_{xxx}\,(l,t) + u\,(t) + d\,(t) \tag{12}$$

**Remark 1**: For clarity and consistency throughout the article, the following symbols are introduced and used: $\frac{\partial(*)}{\partial x} = (*)_x$, $\frac{\partial^2(*)}{\partial x^2} = (*)_{xx}$, $\frac{\partial^3(*)}{\partial x^3} = (*)_{xxx}$, $\frac{\partial(*)}{\partial t} = (*)^{\cdot}$, $\frac{\partial^2(*)}{\partial t^2} = (*)^{\cdot\cdot}$, $\frac{\partial^3(*)}{\partial t^3} = (*)^{\cdot\cdot\cdot}$.

**Assumption 1**: The external disturbances $d(t)$ represent all disturbances acting on the flexible spacecraft and are assumed to be bounded within certain limits. These disturbances, including their derivatives $\dot{d}(t)$, are considered to have an upper bound in magnitude, meaning that $|d(t)| \le \Re$ and $|\dot{d}(t)| \le d_m$, where $\Re$ and $d_m$ are finite values that can be determined based on system specifications and environmental factors.

**Assumption 2**: The $\Sigma_k$ and $\Sigma_p$ of the flexible spacecraft system are assumed to be bounded. For $t > 0, \forall t \in [0, l)$, $\varphi = 1, 2, \varsigma = 2, 3$, $\frac{\partial^{\varphi+1}\eta(x,t)}{\partial t \partial x^\varphi}$ and $\frac{\partial^\varsigma \eta(x,t)}{\partial x^\varsigma}$ are assumed to be bounded.

## 2.3 Control objectives

The objective of this paper is to develop a controller $u(t)$ aimed at mitigating the deformation of the flexible hose $\eta(x,t)$ caused by external disturbances and input constraints. Using the backstepping method, a boundary control law is formulated for the flexible spacecraft system. Furthermore, the closed-loop stability of the system is guaranteed by applying the Lyapunov direct method, along with the careful selection of control parameters to ensure robustness and performance.

## 3 Anti-disturbance boundary controller design

**Lemma 1** (see [37]): For any $\hbar \in R$ and $\ell > 0$, the following inequality holds:

$$0 \le |\hbar| - \hbar \tanh\left(\frac{\hbar}{\ell}\right) \le \upsilon\ell \tag{13}$$

where $\upsilon = 0.2785$.

**Lemma 2** (see [38,39]): For $L(\cdot)$ and $\aleph(\cdot)$ are smooth functions defined on $[0, t_s)$ which $L(t) \ge 0, \forall t \in [0, t_s)$ and $H(\aleph)$ is an even smooth Nussbaum function. Then, the following inequalities exist

$$L \le L(0)e^{-Mt}b_0 + \frac{C}{M}\left(1 - e^{-Mt}\right) + \frac{e^{-Mt}}{Y_\aleph}\int_0^t \left(\varsigma H(\varsigma)\dot{\aleph} - \dot{\aleph}\right)d\tau \tag{14}$$

where $C > 0, M > 0, Y_\aleph > 0$. For $\forall t \in [0, t_s)$, $L(\cdot)$ and $\aleph(\cdot)$ are bounded, then the following inequality holds

$$\zeta = \frac{\partial f(\nu)}{\partial \nu} = \frac{4}{\left(e^{\nu/uC} + e^{-\nu/uC}\right)^2} > 0 \tag{15}$$

**Lemma 3**: Let $\wp_1(x,t), \wp_2(x,t) \in R$ with $\forall (x,t) \in [0,l] \times [0, \infty]$ the following inequalities hold

$$\wp_1(x,t)\wp_2(x,t) \le \frac{1}{f}\wp_1^2(x,t) + f\wp_2^2(x,t) \tag{16}$$

$$\wp_1(x,t)\wp_2(x,t) \le |\wp_1(x,t)\wp_2(x,t)| \le \wp_1^2(x,t) + \wp_2^2(x,t) \tag{17}$$

where $f$ is a constant and satisfies $f > 0$.

**Lemma 4** : For any $\Omega(x, t)$ continuously differentiable on $[\Lambda_1, \Lambda_2]$, the following inequalities hold

$$\int_{\Lambda_1}^{\Lambda_2} (\Omega(x,t))^2 dx \leq 2(\Lambda_2 - \Lambda_1) \Omega^2(\Lambda_2, t) + 4(\Lambda_2 - \Lambda_1)^2 \int_{\Lambda_1}^{\Lambda_2} (\Omega_x(x,t))^2 dx \qquad (18)$$

**Remark 2**: In practical engineering, the influence of disturbances on the system is inevitable. Similarly, there are multiple disturbances $d(t)$ in the flexible spacecraft system. Compared with [40,41], a DO is constructed in this section to estimate the disturbances $d(t)$ in the flexible spacecraft system.

## 3.1 Design of disturbance observer

In this section, a DO is designed to estimate the disturbances $d(t)$ in the flexible spacecraft system. The estimated disturbance is denoted as $\hat{d}(t)$, and the derivative of the external disturbance estimation is given by [42]

$$\dot{\hat{d}}(t) = \Gamma\left(d(t) - \hat{d}(t)\right) \qquad (19)$$

where $\Gamma > 0$.
Next, design an auxiliary function as

$$\beta(t) = \hat{d}(t) - \Gamma m \eta'(l,t) \qquad (20)$$

For the Eq (12), it follows that

$$d(t) = m\eta''(l,t) - \Theta\eta_{xxx}(l,t) - u(t) \qquad (21)$$

Computing the time derivative of Eq (20)

$$\dot{\beta}(t) = \dot{\hat{d}}(t) - \Gamma m\eta''(l,t) = \Gamma\left(d(t) - \hat{d}(t)\right) - \Gamma m\eta''(l,t) \qquad (22)$$

Submitting Eq (21) into Eq (22), then have

$$\begin{aligned} \dot{\beta}(t) &= \Gamma\left(d(t) - \hat{d}(t)\right) - \Gamma m\eta''(l,t) \\ &= \Gamma\left(m\eta''(l,t) - \Theta\eta_{xxx}(l,t) - u(t) - \hat{d}(t)\right) - \Gamma m\eta''(l,t) \\ &= \Gamma\left(-\Theta\eta_{xxx}(l,t) - u(t)\right) - \Gamma\hat{d}(t) \end{aligned} \qquad (23)$$

Noting Eqs (20) and (23), the dynamics of the disturbance observer can express as follows

$$\begin{cases} \hat{d}(t) = \beta(t) + \Gamma m\eta'(l,t) \\ \dot{\beta}(t) = \Gamma\left(-\Theta\eta_{xxx}(l,t) - u(t)\right) - \Gamma\hat{d}(t) \end{cases} \qquad (24)$$

Defining

$$\tilde{d}(t) = d(t) - \hat{d}(t) \qquad (25)$$

where $\tilde{d}(t)$ is the disturbance estimation error.

### 3.2 Design of backstepping boundary controller

In general, as a backstepping control method, the following model transformations can be performed

$$s_1 = x_1 = \eta\,(l, t) \tag{26}$$

$$s_2 = x_2 - \varepsilon_1 = \dot{\eta}\,(l, t) - \varepsilon_1 \tag{27}$$

$$s_3 = u_m\,(u_0\,(t)) - \varepsilon_2 \tag{28}$$

where $\varepsilon_1$ and $\varepsilon_2$ are the virtual control laws to be designed. $u_m$ is a known bound of $u(t)$, $u_0\,(t)$ is the control signal to be designed next. As mentioned above, the input constraint model is described

$$u\,(t) = u_m \tanh\left(\frac{u_0\,(t)}{u_m}\right) \tag{29}$$

**Step 1**: Select the candidate functions of Lyapunov function as follows

$$V_{a1} = \frac{1}{2}s_1^2 \tag{30}$$

The derivative along the trajectory of the Eq (30) can be obtained

$$\dot{V}_{a1} = s_1\dot{s}_1 = s_1\,(s_2 + \varepsilon_1) \tag{31}$$

Select virtual control law $\varepsilon_1$ as

$$\varepsilon_1 = -\frac{r_1}{r_2}s_1 \tag{32}$$

where the selected $r_1$ and $r_2$ are both positive.

Substituting Eq (32) into Eq (31), it can obtain

$$\dot{V}_{a1} = s_1\dot{s}_1 = s_1\left(s_2 - \frac{r_1}{r_2}s_1\right) = -\frac{r_1}{r_2}s_1^2 + s_1 s_2 \tag{33}$$

**Step 2**: Then we choose the candidate functions of Lyapunov function $V_{a2}$ as

$$V_{a2} = V_{a1} + \frac{1}{2}mr_2 s_2^2 \tag{34}$$

The derivative along the trajectory of the Eq (34) with respect to time yields

$$\begin{aligned}
\dot{V}_{a2} &= \dot{V}_{a1} + mr_2 s_2\dot{s}_2 \\
&= -\frac{r_1}{r_2}s_1^2 + s_1 s_2 + mr_2 s_2\,(\eta(l,t)" - \dot{\varepsilon}_1) \\
&= -\frac{r_1}{r_2}s_1^2 + s_1 s_2 + r_2 s_2 m\eta(l,t)" - m\dot{\varepsilon}_1 \\
&= -\frac{r_1}{r_2}s_1^2 + s_1 s_2 + r_2 s_2\,(\Theta\eta_{xxx}(l,t) + s_3 + \varepsilon_2 + d(t) - m\dot{\varepsilon}_1)
\end{aligned} \tag{35}$$

Select virtual control law $\varepsilon_2$ as

$$\varepsilon_2 = m\dot{\varepsilon}_1 - (\kappa_1 + \Delta r_2) s_2 - \frac{s_1}{r_2} \tag{36}$$

where, $\kappa_1 > 0$ and $\Delta > 0$.

Substituting Eq (36) into Eq (35)

$$\begin{aligned}
\dot{V}_{a2} &= -\frac{r_1}{r_2}s_1^2 + s_1 s_2 + r_2 s_2 \left( \Theta\eta_{xxx}(l,t) + s_3 - (\kappa_1 + \Delta r_2) s_2 - \frac{s_1}{r_2} + d(t) \right) \\
&= -\frac{r_1}{r_2}s_1^2 - \kappa_1 r_2 s_2^2 + r_2 s_2 s_3 - \Delta r_2^2 s_2^2 + r_2 s_2 \Theta\eta_{xxx}(l,t) + r_2 s_2 d(t)
\end{aligned} \tag{37}$$

From the following inequality, Eq (37) can be rewritten as

$$s_2 d(t) \le \Delta s_2^2 + \frac{1}{4\Delta}d^2(t) \tag{38}$$

$$\dot{V}_{a2} \le -\frac{r_1}{r_2}s_1^2 - \kappa_1 r_2 s_2^2 + r_2 s_2 s_3 + r_2 s_2 \Theta\eta_{xxx}(l,t) + \frac{1}{4\Delta}d^2(t) \tag{39}$$

Then combining Eqs (12), (26) and (27), and we can obtain

$$\begin{aligned}
m\dot{\varepsilon}_2 &= m\frac{\partial\varepsilon_2}{\partial x_1} + m\frac{\partial\varepsilon_2}{\partial x_2}\eta"(l,t) \\
&= m\frac{\partial\varepsilon_2}{\partial x_1} + m\frac{\partial\varepsilon_2}{\partial x_2}(\Theta\eta_{xxx}(l,t) + u(t) + d(t))
\end{aligned} \tag{40}$$

From Eqs (26)–(29), an auxiliary equation of the controller is designed as

$$\dot{u}_0(t) = \varpi - \psi u_0(t) \tag{41}$$

where the parameter $\psi > 0$, $\varpi$ is the designed auxiliary control law

Differentiating Eq (28)

$$\dot{s}_3 = \frac{\partial u_m(u_0(t))}{\partial u_0(t)}\dot{u}_0(t) - \dot{\varepsilon}_2 \tag{42}$$

Substituting Eq (41) into Eq (42), it can be obtained

$$\begin{aligned}
m\dot{s}_3 &= m\frac{\partial u_m(u_0(t))}{\partial u_0(t)}(\varpi - \psi u_0(t)) - m\dot{\varepsilon}_2 \\
&= m\mathfrak{I}(\varpi - \psi u_0(t)) - m\dot{\varepsilon}_2
\end{aligned} \tag{43}$$

According to Lemma 2, then, it can be obtained

$$\mathfrak{I} = \frac{\partial u_m(u_0(t))}{\partial u_0(t)} = \frac{4}{\left(e^{u_0(t)/u_m(t)} + e^{-u_0(t)/u_m(t)}\right)^2} > 0 \tag{44}$$

**Remark 3**: It is important to highlight that $\mathfrak{I}$ represents an unknown, nonlinear, and time-varying function, which introduces significant challenges in both its analysis and the design of effective controllers. To address the issue of the unknown time-varying function in nonlinear system control, we draw inspiration from the works [20,43–45] . Specifically, we introduce a Nussbaum function $H(\aleph)$ to tackle this complex problem, leveraging its unique properties to effectively manage the uncertainties and variations in the system dynamics.

According to the [43], a Nussbaum function $H(\aleph)$ is even and differentiable and satisfies the properties:

$$\lim_{y \to \infty} \sup \int_0^y H(s)ds = \infty \tag{45}$$

$$\lim_{y \to \infty} \inf \int_0^y H(s)ds = -\infty \tag{46}$$

Then, the auxiliary control law $\varpi$ is designed as

$$\varpi = H(\aleph)\bar{\varpi} \tag{47}$$

Next, a Nussbaum function $H(\aleph)$ is designed as

$$\begin{cases} H(\aleph) = \aleph^2 \cos(\aleph) \\ \dot{\aleph} = \gamma_\aleph m s_3 \bar{\varpi} \end{cases} \tag{48}$$

where $\gamma_\aleph$ is a design parameter and satisfies $\gamma_\aleph > 0$.
Then, we design $\bar{\varpi}$ as

$$\begin{aligned} \bar{\varpi} = {} & \frac{\partial \varepsilon_2}{\partial x_1} x_2 + \frac{1}{m} \frac{\partial \varepsilon_2}{\partial x_2} u(t) - \frac{b_1}{m} s_3 + \mathfrak{I}\psi u_0(t) - \frac{r_2}{m} s_2 \\ & - b_2 \left( \frac{\partial \varepsilon_2}{\partial x_2} \right)^2 s_3 + \frac{1}{m} \frac{\partial \varepsilon_2}{\partial x_2} \Theta \eta_{xxx}(l, t) \end{aligned} \tag{49}$$

where the parameters $b_1 > 0$ and $b_2 > 0$.
Substituting Eqs (40), (43) into Eq (49), one has

$$\begin{aligned} m\dot{s}_3 + m\bar{\varpi} = {} & \frac{\partial u_m(u_0(t))}{\partial u_0(t)} (\varpi - \psi u_0(t)) - m\dot{\varepsilon}_2 + m\frac{\partial \varepsilon_2}{\partial x_1} x_2 + \frac{\partial \varepsilon_2}{\partial x_2} u(t) - b_1 s_3 \\ & + m\mathfrak{I}\psi u_0(t) - r_2 s_2 - mb_2 \left( \frac{\partial \varepsilon_2}{\partial x_2} \right)^2 s_3 + \frac{\partial \varepsilon_2}{\partial x_2} \Theta \eta_{xxx}(l, t) \\ = {} & m\mathfrak{I}\varpi - mb_2 \left( \frac{\partial \varepsilon_2}{\partial x_2} \right)^2 s_3 - b_1 s_3 - r_2 s_2 - \frac{\partial \varepsilon_2}{\partial x_2} d(t) \end{aligned} \tag{50}$$

**Step 3**: Consider the following Lyapunov function $V_a(t)$ as

$$V_a(t) = V_{a2}(t) + \frac{1}{2} m s_3^2 + \frac{1}{2} \tilde{d}^2(t) \tag{51}$$

The derivative along the trajectory of the Eq (51)

$$\dot{V}_a(t) = \dot{V}_{a2}(t) + ms_3\dot{s}_3 + \tilde{d}(t)\dot{\tilde{d}}(t)$$
$$\leq -\frac{r_1}{r_2}s_1^2 - \kappa_1 r_2 s_2^2 + r_2 s_2 s_3 + r_2 s_2 \Theta \eta_{xxx}(l,t) \tag{52}$$
$$+ \frac{1}{4\Delta}d^2(t) + ms_3\dot{s}_3 + \tilde{d}(t)\left(\dot{d}(t) - \dot{\hat{d}}(t)\right)$$

Substituting Eqs (47) and (50) into Eq (52), it can be further obtained

$$\dot{V}_a(t) \leq -\frac{r_1}{r_2}s_1^2 - \kappa_1 r_2 s_2^2 + r_2 s_2 \Theta \eta_{xxx}(l,t)$$
$$+ s_3(m\dot{s}_3 + m\varpi) - ms_3\varpi + \tilde{d}(t)\left(\dot{d}(t) - \dot{\hat{d}}(t)\right)$$
$$+ \frac{1}{4\Delta}d^2(t)$$
$$\leq -\frac{r_1}{r_2}s_1^2 - \kappa_1 r_2 s_2^2 + r_2 s_2 \Theta \eta_{xxx}(l,t)$$
$$+ s_3\left(m\mathfrak{I}\varpi - mb_2\left(\frac{\partial\varepsilon_2}{\partial x_2}\right)^2 s_3 - b_1 s_3 - r_2 s_2 - \frac{\partial\varepsilon_2}{\partial x_2}d(t)\right)$$
$$- ms_3\varpi + \frac{1}{4\Delta}d^2(t) \tag{53}$$
$$+ \tilde{d}(t)\left(\dot{d}(t) - \dot{\hat{d}}(t)\right)$$
$$\leq -\frac{r_1}{r_2}s_1^2 - \kappa_1 r_2 s_2^2 - b_1 s_3^2 + r_2 s_2 \Theta \eta_{xxx}(l,t)$$
$$- mb_2\left(\frac{\partial\varepsilon_2}{\partial x_2}\right)^2 s_3^2 - s_3\frac{\partial\varepsilon_2}{\partial x_2}d(t)$$
$$+ \frac{1}{4\Delta}d^2(t) + \tilde{d}(t)\left(\dot{d}(t) - \dot{\hat{d}}(t)\right)$$
$$+ ms_3\varpi(\mathfrak{I}H(\aleph) - 1)$$

Combining inequalities Eqs (12), (20)–(22), (25) and (38), it is noted that

$$-mb_2\left(\frac{\partial\varepsilon_2}{\partial x_2}\right)^2 s_3^2 - s_3\frac{\partial\varepsilon_2}{\partial x_2}d(t) \leq \frac{1}{4mb_2}d^2(t) \tag{54}$$

$$V_d(t) = \tilde{d}(t)\left(\dot{d}(t) - \dot{\hat{d}}(t)\right)$$
$$= -\tilde{d}(t)\left(\dot{\beta}(t) + \Gamma m\eta''(l,t)\right) + \tilde{d}(t)\dot{d}(t)$$
$$= -\tilde{d}(t)\left(\Gamma\left(-\Theta\eta_{xxx}(l,t) - u(t)\right) - \Gamma\hat{d}(t) + \Gamma m\eta''(l,t)\right) + \tilde{d}(t)\dot{d}(t) \tag{55}$$
$$= \tilde{d}(t)\dot{d}(t) - \Gamma\tilde{d}^2$$

According to Lemma 3 and assumption 1, Eq (55) follows that

$$
\begin{aligned}
V_d(t) &= \tilde{d}(t)\dot{d}(t) - \Gamma\tilde{d}^2 \\
&\leq -\Gamma\tilde{d}^2(t) + \mu\tilde{d}^2(t) + \frac{1}{\mu}\dot{d}^2(t) \\
&\leq \frac{1}{\mu}d_m^2 + \tilde{d}^2(t)(\mu - \Gamma)
\end{aligned}
\tag{56}
$$

Therefore

$$
\begin{aligned}
\dot{V}_a(t) &\leq -\frac{r_1}{r_2}s_1^2 - \kappa_1 r_2 s_2^2 - b_1 s_3^2 + r_2 s_2 \Theta\eta_{xxx}(l,t) - \frac{1}{4mb_2}d^2(t) + \frac{1}{4\Delta}d^2(t) + \frac{1}{\mu}d_m^2 \\
&\quad + \tilde{d}^2(t)(\mu - \Gamma) + \frac{1}{\gamma_\aleph}(\mathfrak{J}H(\aleph) - 1)\dot{\aleph}
\end{aligned}
\tag{57}
$$

**Theorem 1**: With the dynamic equation of flexible spacecraft based on PDE (see Eq (12)) and boundary conditions (see Eqs (10) and (11)), under the proposed control law (see Eqs (32), (36), (47) and (49)), then the following properties hold:

1. The closed-loop system (Eq (12)) is uniformly bounded, and $\|\eta(x,t)\| \leq \Omega_R$, where $\Omega_R = \sqrt{\frac{\beta V_{all}(t)}{4r_2\Theta q_2}}$.

2. The control input (Eq (12)) is bounded, and have $|u_0(t)| = u_m\left|\tanh\left(\frac{u_0(t)}{u_m}\right)\right| \leq u_m$.

**Remark 4**: The design of the control law relies on the Lyapunov direct method, with the assumption that all signals involved in the boundary controller and disturbance observer (as described in Eq 24) can either be directly measured by sensors or derived through a backward difference algorithm. Specifically, the values of $\eta(l,t)$ and $\eta_x(l,t)$ can be obtained using a laser displacement sensor and an inclinometer placed at the top boundary of the flexible spacecraft. Additionally, the backward difference algorithm allows for the computation of $\eta_x(l,t)$ and $\eta(l,t)$ from the available measurements. However, in the practical docking scenario of the flexible spacecraft, there will inevitably be discrepancies between the sensor measurements and the actual system behavior. These measurement errors can influence the controller's performance. Therefore, when designing the controller, it is crucial to carefully select the appropriate control parameters to ensure satisfactory performance despite these uncertainties.

**Proof**: Considering the following Lyapunov function

$$
V_{all}(t) = V_a(t) + V_b(t) + V_c(t)
\tag{58}
$$

where

$$
V_b(t) = \frac{r_2}{2}\rho\int_0^l (\eta'(x,t))^2 dx + \frac{r_2}{2}\Theta\int_0^l (\eta_{xx}(x,t))^2 dx + \frac{r_1}{2}\Gamma_1\int_0^l (\eta(x,t))^2 dx
\tag{59}
$$

$$
V_c(t) = r_1\rho\int_0^l (\eta'(x,t)\eta(x,t))\,dx
\tag{60}
$$

Take the derivation of Eq (59), it follows that

$$\dot{V}_b(t) = r_2\rho \int_0^l \eta'(x,t)\,\eta''(x,t)\,dx + r_2\Theta \int_0^l \eta_{xx}(x,t)\,\eta_{xxt}(x,t)\,dx + r_1\Gamma_1 \int_0^l \eta(x,t)\,\eta'(x,t)\,dx \tag{61}$$

Substituting Eq (10) and boundary conditions into the Eq (61), $\dot{V}_b(t)$ can be rewritten as

$$\dot{V}_b(t) = -r_2\Gamma_1 \int_0^l (\eta'(x,t))^2 dx + r_1\Gamma_1 \int_0^l \eta(x,t)\,\eta'(x,t)\,dx - r_2\eta'(l,t)\,(\Theta\eta_{xxx}(l,t)) \tag{62}$$

Computing the time derivative along the trajectory of the Eq (60)

$$\dot{V}_c(t) = r_1\rho \int_0^l (\eta''(x,t)\,\eta(x,t))\,dx + r_1\rho \int_0^l (\eta'(x,t))^2 dx \tag{63}$$

Substituting Eq (10) and boundary conditions into the Eq (63), then use integration by parts, we have

$$
\begin{aligned}
\dot{V}_c(t) &= r_1\rho \int_0^l \left(\frac{-\Gamma_1\eta'(x,t) - \Theta\eta_{xxxx}(x,t)}{\rho}\right)\eta(x,t)\,dx + r_1\rho \int_0^l (\eta'(x,t))^2 dx \\
&= -r_1\Gamma_1 \int_0^l \eta'(x,t)\,\eta(x,t)\,dx - r_1\Theta \int_0^l \eta_{xxxx}(x,t)\,\eta(x,t)\,dx \\
&\quad + r_1\rho \int_0^l (\eta'(x,t))^2 dx \\
&= r_1\rho \int_0^l (\eta'(x,t))^2 dx - r_1\Gamma_1 \int_0^l \eta'(x,t)\,\eta(x,t)\,dx \\
&\quad - r_1\Theta \int_0^l (\eta_{xx}(x,t))^2 dx - r_1\Theta\eta(l,t)\,\eta_{xxx}(l,t)
\end{aligned}
\tag{64}
$$

From Eqs (56), (62) and (64), we further obtain the Lyapunov function $\dot{V}_{all}(t)$

$$
\begin{aligned}
\dot{V}_{all}(t) &\leq -\frac{r_1}{r_2}s_1^2 - \kappa_1 r_2 s_2^2 - b_1 s_3^2 - \frac{1}{4mb_2}d^2(t) + \frac{1}{4\Delta}d^2(t) + \frac{1}{\mu}d_m^2 \\
&\quad + \tilde{d}^2(t)(\mu - \Gamma) + \frac{1}{\gamma_\aleph}(\mathfrak{J}H(\aleph) - 1)\dot{\aleph} \\
&\quad - r_2\Gamma_1 \int_0^l (\eta'(x,t))^2 dx - r_2\eta'(l,t)\,(\Theta\eta_{xxx}(l,t)) \\
&\quad + r_1\rho \int_0^l (\eta'(x,t))^2 dx - r_1\Theta \int_0^l (\eta_{xx}(x,t))^2 dx \\
&\leq -\frac{r_1}{r_2}s_1^2 - \kappa_1 r_2 s_2^2 - b_1 s_3^2 - \left(\frac{1}{4mb_2} - \frac{1}{4\Delta}\right)\mathfrak{R}^2 \\
&\quad + \frac{1}{\mu}d_m^2 + \tilde{d}^2(t)(\mu - \Gamma) + \frac{1}{\gamma_\aleph}(\mathfrak{J}H(\aleph) - 1)\dot{\aleph} \\
&\quad - (r_2\Gamma_1 - r_1\rho) \int_0^l (\eta'(x,t))^2 dx - r_1\Theta \int_0^l (\eta_{xx}(x,t))^2 dx
\end{aligned}
\tag{65}
$$

According to Lemma 4, then we can get the following inequality

$$-\Lambda_1 r_3 (\eta\,(x,t))^2 \le -\Lambda_1 \int_0^l \eta(x,t)^2 dx + \Lambda_1 r_3^4 \int_0^l \eta_{xx}(x,t)^2 dx \tag{66}$$

where $\Lambda_1$ and $r_3$ are constants, and $\Lambda_1 > 0$, $r_3 > 0$.

$$
\begin{aligned}
\dot{V}_{all}\,(t) \le & -\left(\frac{r_1}{r_2} - \Lambda_1 r_3\right) s_1^2 - \kappa_1 r_2 s_2^2 - b_1 s_3^2 \\
& + \left(\frac{1}{4\Delta} - \frac{1}{4mb_2}\right) \mathfrak{R}^2 + \frac{1}{\mu} d_m^2 + \tilde{d}^2\,(t)\,(\mu - \Gamma) \\
& + \frac{1}{\gamma_\aleph} (\mathfrak{I} H\,(\aleph) - 1)\dot{\aleph} - (r_2\Gamma_1 - r_1\rho) \int_0^l (\eta\,(x,t))^2 dx \\
& - \left(r_1\Theta - \Lambda_1 r_3^4\right) \int_0^l (\eta_{xx}\,(x,t))^2 dx - \Lambda_1 \int_0^l \eta(x,t)^2 dx
\end{aligned}
\tag{67}
$$

From Eq (59), the Lyapunov function $V_b\,(t)$ is bounded as

$$V_b\,(t) \ge \Pi_1 \left( \int_0^l (\eta\,(x,t))^2 dx + \int_0^l (\eta\,(x,t))^2 dx \right) \tag{68}$$

where $\Pi_1 = \min\left(\frac{r_2\rho}{2}, \frac{\Gamma_1 r_1}{2}\right) > 0$.

Similarly, from Eq (60), the Lyapunov function $V_c\,(t)$ is bounded as

$$\|V_c\,(t)\| \le r_1\rho \int_0^l (\eta\,(x,t)\,\eta\,(x,t))\,dx \le \Pi_2 V_b\,(t) \tag{69}$$

where $\Pi_2 = \frac{r_1\rho}{\Pi_1}$.

Then

$$0 \le \Pi_4 V_b\,(t) \le V_b\,(t) + V_c\,(t) \le \Pi_3 V_b\,(t) \tag{70}$$

where $\Pi_3 = 1 + \Pi_2 > 1$, $\Pi_4 = 1 - \Pi_2 < 1$.

Combining with Eq (58), we can obtain

$$q_1\,(V_a\,(t) + V_b\,(t)) \le V_{all}\,(t) \le q_2\,(V_a\,(t) + V_b\,(t)) \tag{71}$$

where $q_1 = \max\,(\Pi_3, 1) = \Pi_3$, $q_2 = \min\,(\Pi_4, 1) = \Pi_4$.

Considering the Lyapunov equation (67), selecting appropriate parameters $r_1, r_2, r_3, \kappa_1, \Lambda_1$ and $b_1$ can satisfy the following conditions:

$$\Phi_1 = \frac{r_1}{r_2} - \Lambda_1 r_3 > 0 \tag{72}$$

$$\Phi_2 = b_1 > 0 \tag{73}$$

$$\Phi_3 = \kappa_1 r_2 > 0 \tag{74}$$

$$\Phi_4 = \mu - \Gamma < 0 \tag{75}$$

$$\Phi_5 = r_2 \Gamma_1 - r_1 \rho > 0 \tag{76}$$

$$\Phi_6 = r_1 \Theta - \Lambda_1 r_3^4 > 0 \tag{77}$$

From the Eq (71), the Eq (67) $\dot{V}_{all}(t)$ can be further rewritten as

$$\dot{V}_{all}(t) \le -q_3 \left( V_a(t) + V_b(t) \right) + \frac{1}{\mu} d_m^2 + \frac{1}{\gamma_\aleph} (\Im H(\aleph) - 1) \dot{\aleph} + \left( \frac{1}{4\Delta} - \frac{1}{4mb_2} \right) \mathfrak{R}^2 \tag{78}$$

where $q_3 = 2 \min \left( \frac{\Phi_1}{mr_1} \frac{\Phi_2}{m}, \frac{\Phi_3}{mr_2}, \frac{\Phi_4}{m}, \frac{\Phi_5}{mr_2}, \frac{\Phi_6}{r_2 EI} \right) > 0$.

Following the Eqs (71) and (78) it can be obtained

$$\dot{V}_{all}(t) \le -q_4 V_{all}(t) + \frac{1}{\mu} d_m^2 + \frac{1}{\gamma_\aleph} (\Im H(\aleph) - 1) \dot{\aleph} + \left( \frac{1}{4\Delta} - \frac{1}{4mb_2} \right) \mathfrak{R}^2 \tag{79}$$

where

$$q_4 = \left\{ \frac{q_3}{q_1} > 0 \,\&\, q_4 \ge \max \left( \frac{1}{V_{all}(t)} \left( \frac{1}{\mu} d_m^2 + \frac{1}{\gamma_\aleph} (\Im H(\aleph) - 1) \dot{\aleph} + \left( \frac{1}{4\Delta} - \frac{1}{4mb_2} \right) \mathfrak{R}^2 \right) \right) \right\} \tag{80}$$

Then, by integrating Eq (80), we can obtain

$$V_{all}(t) \le V_{all}(0) e^{-q_4} + \frac{1}{q_4} \left( \left( \frac{1}{4\Delta} - \frac{1}{4mb_2} \right) \mathfrak{R}^2 + \frac{1}{\mu} d_m^2 \right) \left( 1 - e^{-q_4 t} \right) + \frac{e^{-q_4 t}}{\gamma_\aleph} \int_0^t (\Im H(\aleph) - 1) \dot{\aleph} e^{-q_4 t} d\tau \tag{81}$$

From Lemma 2, we can draw a conclusion that $V_{all}(t)$, $\aleph$, $s_1$, $s_2$, $s_3$, $\eta(x,t)$ and $\eta_t(x,t)$ are all bounded on $[0, t)$. Then, based on assumption 2, we can have the following properties:

$$|u_0(t)| = u_m \left| \tanh \left( \frac{u_0(t)}{u_m} \right) \right| \le u_m \tag{82}$$

$$\left| \frac{\partial u_m(u_0)}{\partial u_0} \right| = \left| \frac{4}{\left( e^{u_0/u_m} + e^{-u_0/u_m} \right)^2} \right| \le 1 \tag{83}$$

$$\left| \frac{\partial u_m(u_0)}{\partial u_0} u_0(t) \right| = \left| \frac{4 u_0(t)}{\left( e^{u_0/u_m} + e^{-u_0/u_m} \right)^2} \right| \le 0.5 u_m \tag{84}$$

Moreover, combining the following inequality

$$\int_0^l \left(\eta_{xx}\left(x,t\right)\right)^2 dx \geq \frac{1}{l^2} \int_0^l \left(\eta_x\left(x,t\right)\right)^2 dx \geq \frac{1}{l^3} \left(\eta\left(x,t\right)\right)^2 \qquad (85)$$

Furthermore, it can be obtained that

$$\left\|\eta\left(x,t\right)\right\| \leq \sqrt{\frac{l^3 V_{all}\left(t\right)}{4 r_2 \Theta q_2}} \qquad (86)$$

This completes the proof.

## 4 Simulation Examples

### 4.1 Two control methods for comparison

In this paper, the simulation is based on MATLAB/SIMULINK, with a simulation period of 50s and a simulation step of 0.001 seconds. We have updated the manuscript to include the MATLAB/Simulink version used for the simulations and the specifications of the machine. The simulations were run on a system with an 11th Gen Intel(R) Core(TM) i7-1165G7 processor at 2.80 GHz, 16.0 GB of RAM (15.7 GB usable), Intel(R) Iris(R) Xe Graphics (128 MB), and a 477 GB SSD (Samsung MZALQ512HALU-000L1). The operating system is 64-bit based on an x64 processor.

To better demonstrate the robustness of flexible spacecraft, this paper proposes a boundary control strategy based on the Back-Stepping method (denoted as BCBS). Additionally, a comparison is made with a boundary control strategy based on Proportional-Derivative (PD) control, referred to as the BCPD method. In this simulation, the finite difference method is employed to present the simulation results. The relevant parameters of the flexible spacecraft are provided in Table 1. In addition, the main code program of the paper is shown in the S1 attachment uploaded.

The initial condition of the flexible hose is set as $\eta\left(x,t\right) = 0.18x$. For better showing the superiority of the control method (BCBS) proposed in this paper, the flexible hose parameters and simulation environment in the BCPD method are the same as those in BCBS method. The boundary control strategy based on PD is set as

$$u_{BCPD} = -k_p \eta\left(l,t\right) - m\frac{\alpha_1}{\beta_1}\dot{\eta}\left(l,t\right) - k_d \pi\left(t\right) \qquad (87)$$

where $\pi\left(t\right) = \frac{\alpha_1}{\beta_1}\eta\left(l,t\right) + \dot{\eta}\left(l,t\right)$, $k_p > 0$ and $k_d > 0$ are the control gains, $\alpha_1$ and $\beta_1$ are positive weighting constants. In addition, the control parameters based on BCBS and BCPD methods are shown in Table 2.

Table 1. **Parameters of a flexible spacecraft.**

| Symbols | Description | Value |
| --- | --- | --- |
| $l$ | The length of flexible spacecraft | $10\ m$ |
| $m$ | The mass of flexible spacecraft | $80\ kg$ |
| $\rho$ | Density of hose material | $2.715 \times 10^3\ kg/m^3$ |
| $E$ | Young's modulus | $6.90 \times 10^{10}\ N/m^2$ |
| $\Gamma_1$ | Elastic damping coefficient | 0.0045 |
| $I$ | Area moment of inertia of the flexible spacecraft | $1.70 \times 10^{-7}\ m^4$ |

**Table 2.** Controller parameters of two control methods.

| Control method | BCBS method | BCPD method |
|---|---|---|
| Controller parameters | Controller for the BCBS method:<br>$r_1 = 25$, $r_2 = 1100$, $r_3 = 5$<br>$\gamma_\aleph = 5$, $b_1 = 2$, $\kappa_1 = 40$<br>$\Lambda_1 = 10$, $b_2 = 2$<br>The control input is constrained as:<br>$\left\|u_{BCBS}\right\| \leq 1200N$ | Controller for the BCPD method:<br>$k_p = 12.5$, $k_d = 480$<br>Positive weighting constants:<br>$\alpha_1 = 0.2$, $\beta_1 = 10$<br>The control input is constrained as:<br>$\left\|u_{BCPD}\right\| \leq 800N$ |

## 4.2 Simulation results

To demonstrate that the BCBS method offers greater robustness to external disturbances, this paper considers two types of disturbances: constant disturbance and time-varying disturbance. These disturbances are used to highlight the ability of the BCBS method to effectively handle external influences and maintain system stability.

(a) Constant disturbance : $d(t) = 0.1015$.

(b) Time-varying disturbance: $d(t) = 0.02 + 0.018*\sin(0.5\pi t)$

As depicted in Figs 2, 3 and 4, it can be seen that the vibration deformation $\eta(x, t)$ of flexible spacecraft without control method, BCPD method and BCBS method respectively.

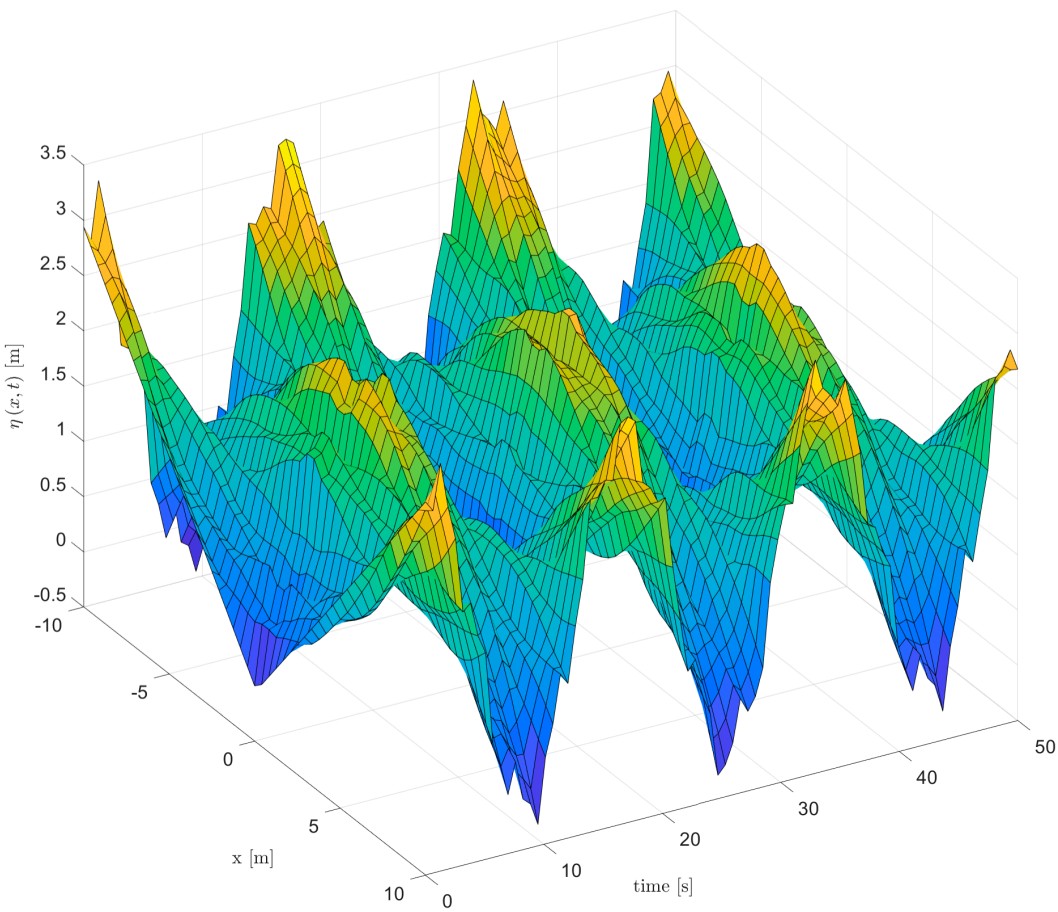

**Fig 2. The elastic deflection of the flexible spacecraft without control.**

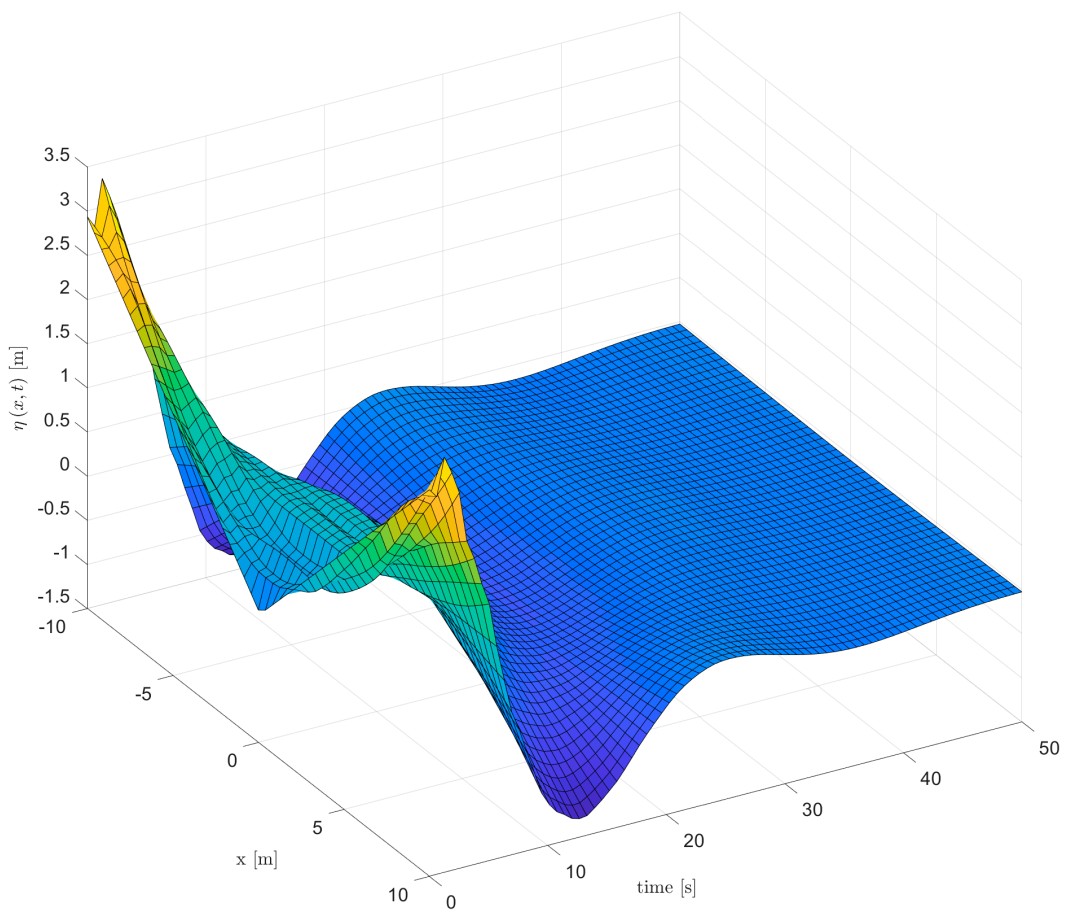

**Fig 3. The elastic deflection of the flexible spacecraft with PD control.**

From Fig 2, it can be seen that the vibration deformation $\eta(x,t)$ of the flexible spacecraft is very obvious without any control. The vibration deformation $\eta(x,t)$ of the flexible spacecraft shown in Fig 3 exhibits a amplitude compared to that in Fig 2 when the BCPD method is applied. Furthermore, the BCBS method proposed in this paper effectively reduces the vibration deformation $\eta(x,t)$ to a negligible level within 35 seconds, with the amplitude notably decreasing during the initial 30 seconds. This demonstrates that the BCBS method achieves rapid and effective attenuation of the lateral vibration $\eta(x,t)$. By comparing the elastic deflections in the three figures, we can see that the proposed control method offers significant advantages in suppressing the elastic vibrations of the spacecraft. Without control, the spacecraft exhibits large vibration amplitudes and long settling times, with the system unable

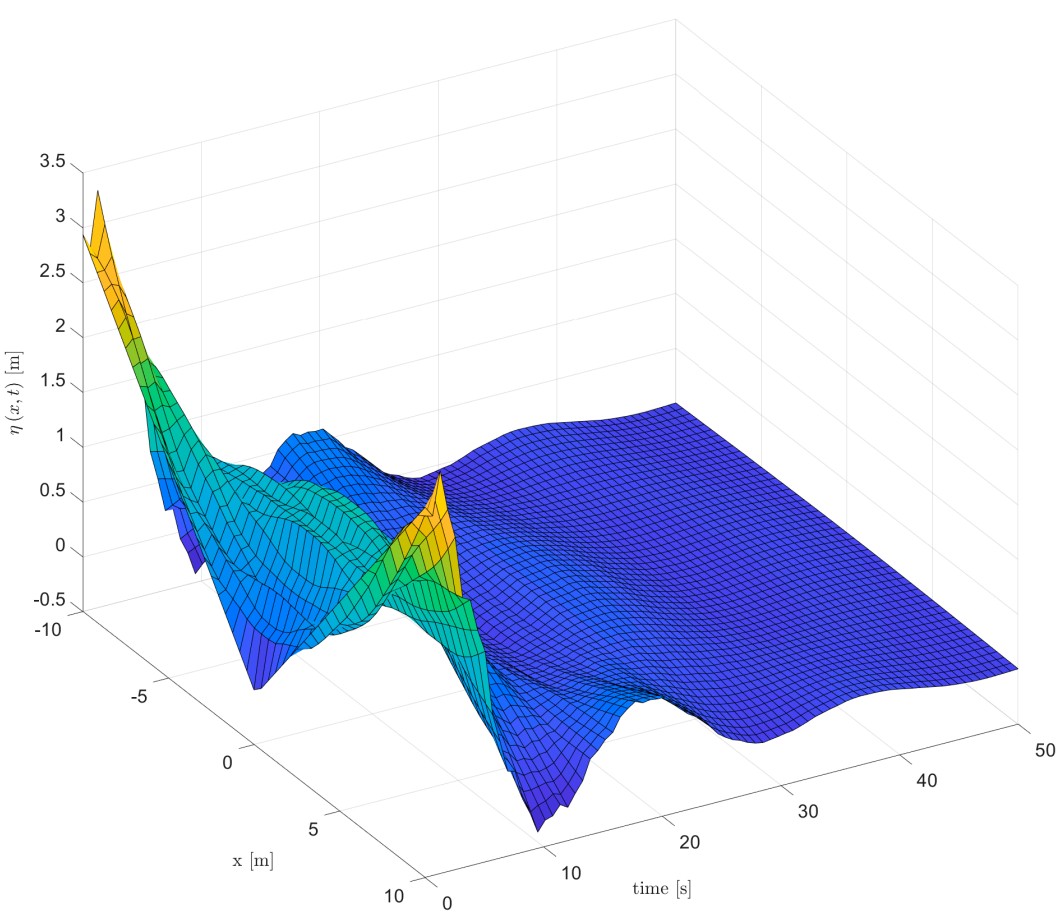

**Fig 4. The elastic deflection of the flexible spacecraft with proposed control.**

to effectively dampen disturbances. With PD control, although the vibrations are reduced, there are still substantial peak overshoots and long settling times. In contrast, the proposed control method quickly attenuates the vibrations to near zero, significantly reducing both peak overshoot and settling time, demonstrating superior vibration suppression performance. Quantitative analysis shows that the proposed method reduces settling time by approximately 80%, peak overshoot by about 94%, and steady-state error to near zero, proving its superiority in suppressing elastic vibrations in spacecraft. From Figs 5 and 6, it can be seen that the boundary displacement of flexible spacecraft $\eta(0, t)$, $\eta(l, t)$ under the control method (BCPD method and BCBS method) and without control. As can be seen from Figs 5 and 6, compared

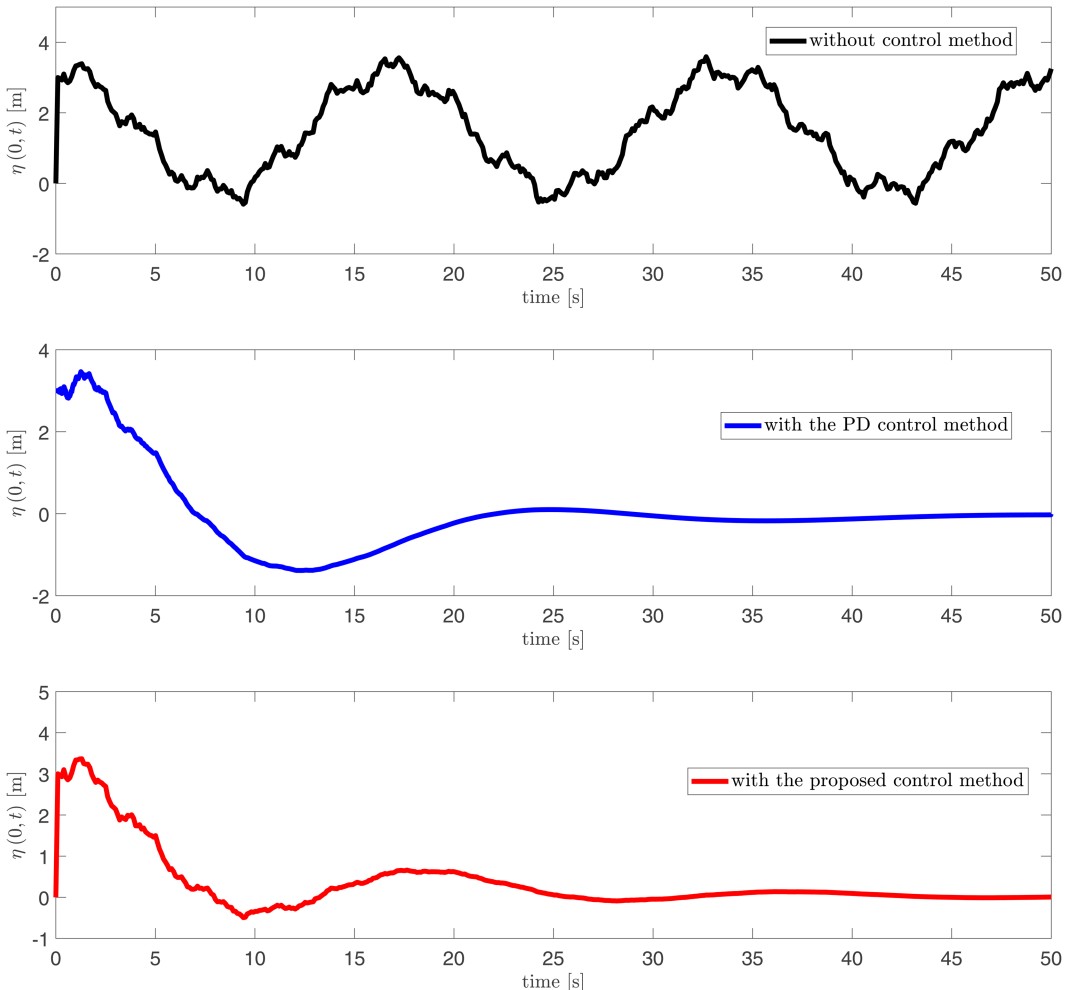

**Fig 5. The boundary displacement of flexible spacecraft $\eta\,(0,t)$ with control and without control.**

with the method without control and BCPD method, the boundary displacement of the flexible spacecraft in the control method proposed (BCBP method) in this paper approaches zero after about 25$s$, which shows the anti-disturbance and robustness of BCBS method are better.

The responses of disturbance $d(t)$, its estimation and disturbance error $e_{d(t)}$ against a constant disturbance in Fig 7. Different from Fig 7, the responses of disturbance $d(t)$, its estimation and disturbance error $e_{d(t)}$ against time-varying disturbance in Fig 8. From Figs 7 and 8, it can be observed that, for both constant and time-varying disturbances, the disturbance observer designed in this paper accurately estimates the disturbance, with the disturbance estimation error $e_{d(t)}$ converging to near zero. These results indicate that the proposed

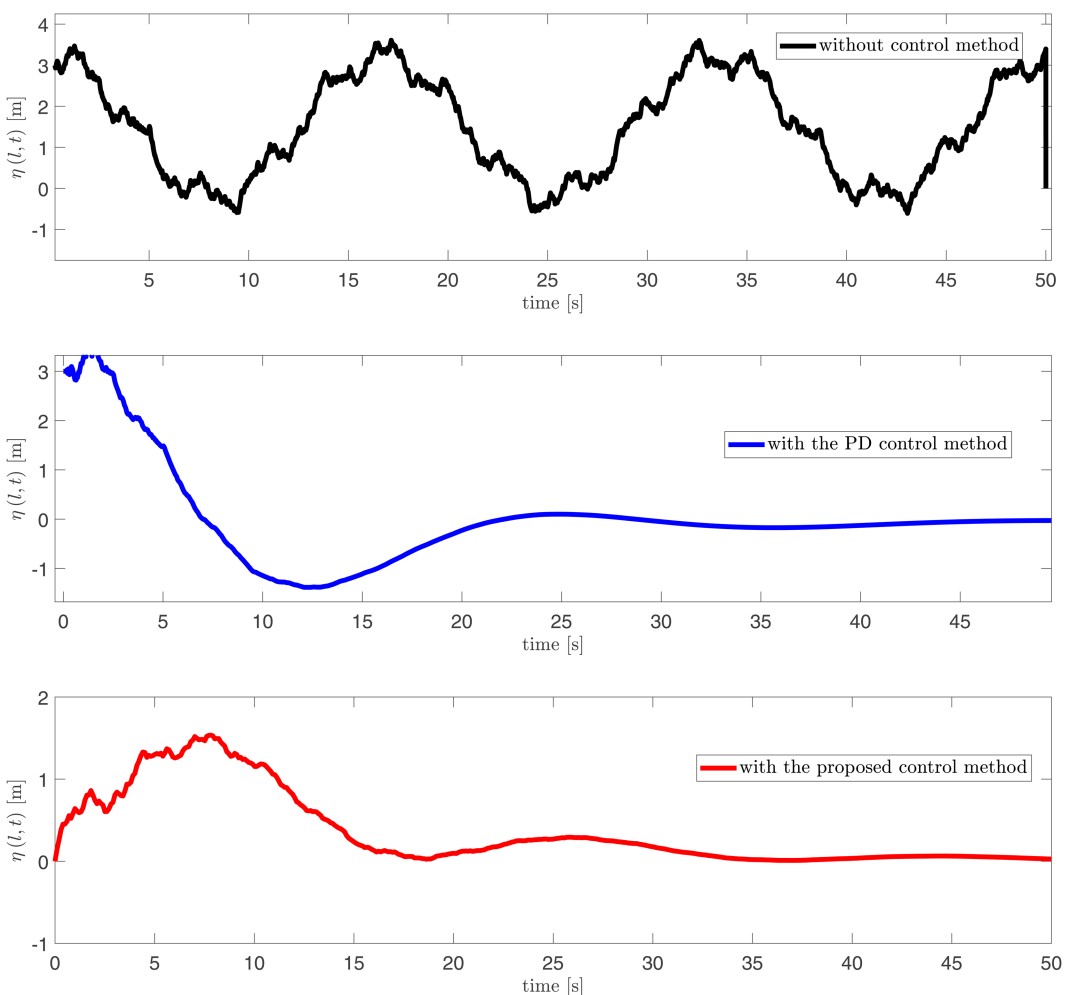

**Fig 6. The boundary displacement of flexible spacecraft $\eta\left(l, t\right)$ with control and without control.**

disturbance observer possesses effective disturbance estimation capability under different disturbance conditions. The response of control input under the BCPD methods and BCBS methods are shown in Fig 9. Fig 9 compares the control inputs under PD control and the proposed control method. The PD control input (top) exhibits large oscillations with peaks reaching up to 1000 N, indicating high control effort and potentially inefficient actuator usage. In contrast, the proposed control (bottom) limits the peaks to about 50 N, representing a 95% reduction in control input magnitude. Additionally, the proposed control shows smoother and more stable input throughout the process, suggesting lower energy consumption and reduced actuator stress. This demonstrates that the proposed control method is significantly more efficient, requiring much less control effort while still achieving effective system stabilization.

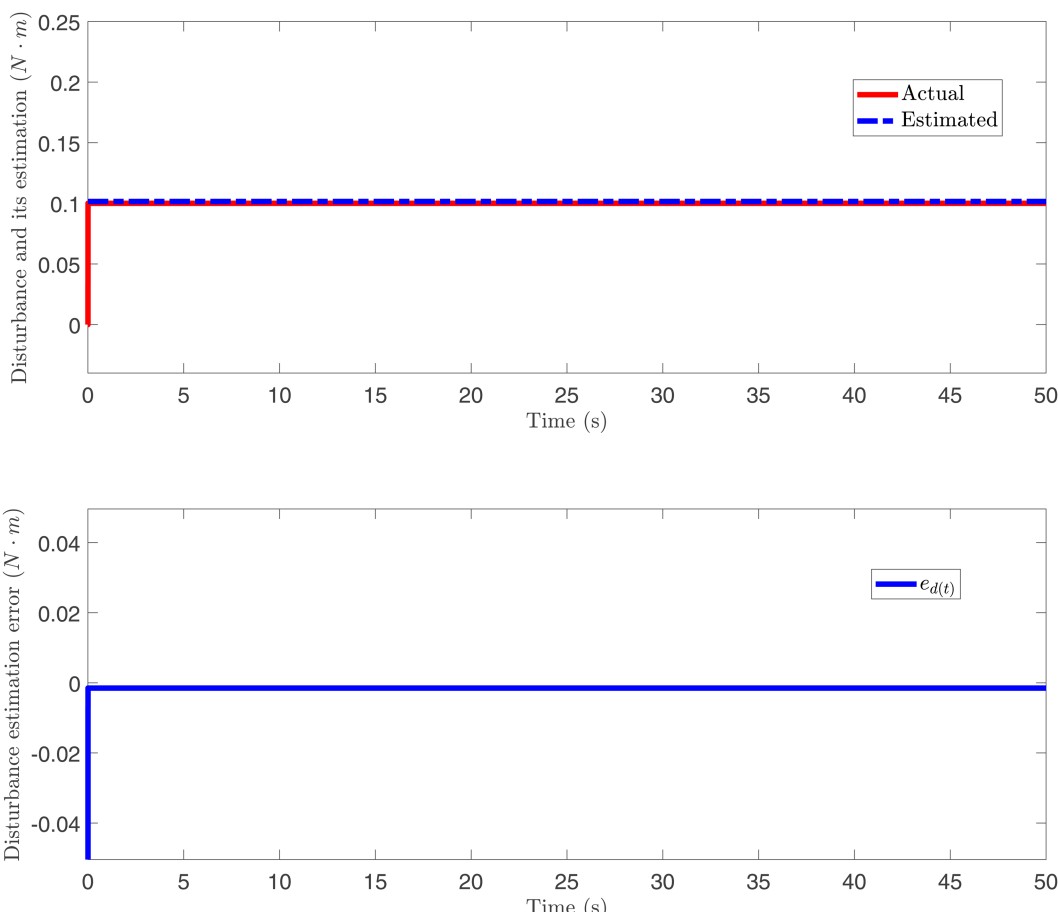

**Fig 7. The responses of disturbance $d(t)$, its estimation and disturbance error against a constant disturbance.**

## 5 Conclusion

In this paper, the problem of the vibration deformation of flexible spacecraft subject to external disturbances and control input constrained are investigated. Based on PDEs and boundary initial conditions, a novel backstepping control strategy based on Nussbaum function and disturbance observer is proposed to suppress the elastic vibration of the flexible spacecraft.

- First, unlike traditional ordinary differential equations (ODEs) that fail to accurately capture the physical characteristics of flexible systems, this paper utilizes PDEs to effectively model the dynamic behavior of flexible spacecraft, providing a more precise description of their characteristics.
- Second, a backstepping control approach is introduced (as referenced in [20,25,31,32]), incorporating a Nussbaum function and a disturbance observer to address the challenges posed by external disturbances and input constraints, ensuring better performance and stability.

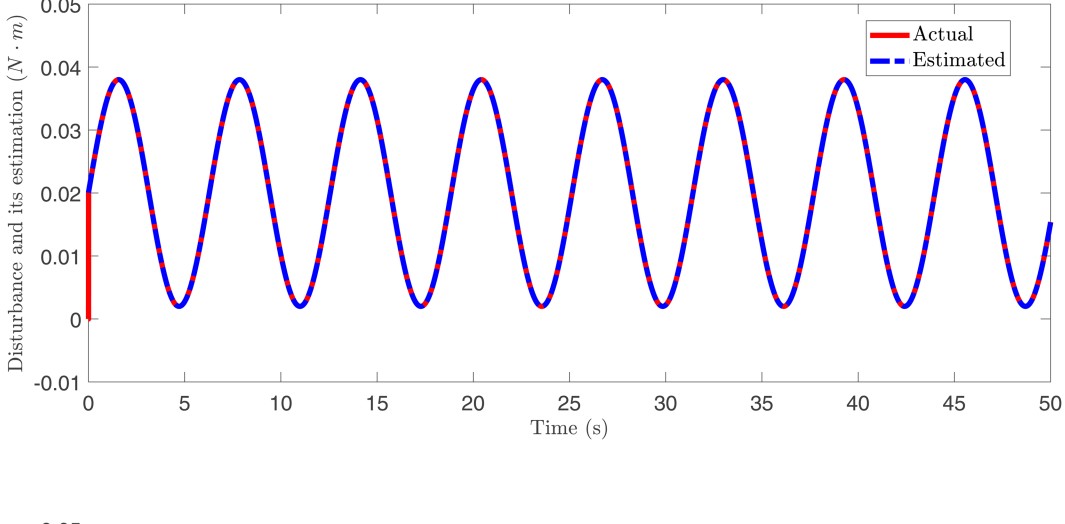

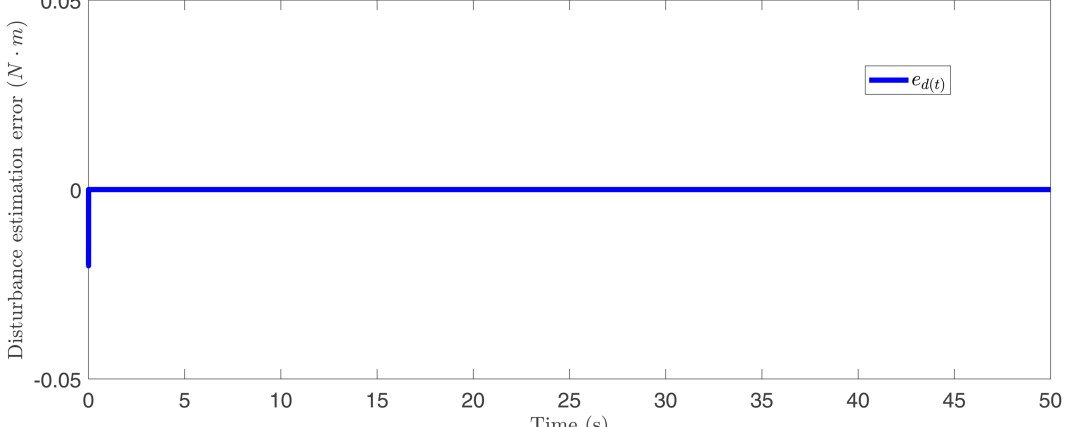

**Fig 8. The responses of disturbance $d(t)$, its estimation and disturbance error against a time-varying disturbance.**

- Last but not least, by applying the Lyapunov direct method and selecting suitable control parameters, the system can be steered toward a compact set, achieving desired performance. Furthermore, numerical simulations demonstrate the effectiveness and robustness of the proposed control strategy, providing further validation of its applicability.

The vibration of flexible spacecraft can significantly impact the success rate of aerial refueling docking operations. Therefore, future research will focus on conducting a comprehensive safety analysis of flexible spacecraft to better understand and mitigate the risks associated with these vibrations during critical operations.

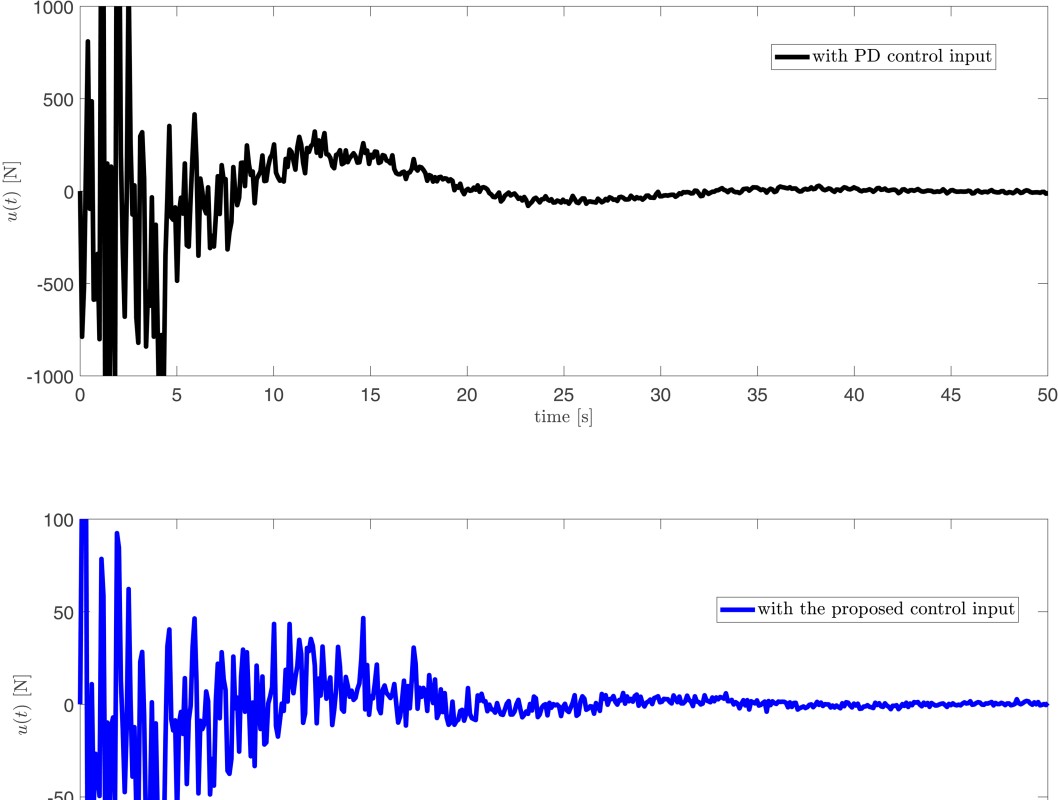

**Fig 9. The diagram of control input under the two control methods.**

## Supporting information

**S1 Text. Paper program.**
(PDF)

## Author contributions

**Data curation:** Bo Zhang, Ming Wen.

**Methodology:** Bo Zhang, Ming Wen.

**Writing – original draft:** Bo Zhang.

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
