## [Decision Letter · Decision Letter 0]

10 Jul 2025

PONE-D-25-18264Active control of flexible spacecraft in orbit based on partial differential equationsPLOS ONE

Dear Dr. Zhang,

Thank you for submitting your manuscript to PLOS ONE. After careful consideration, we feel that it has merit but does not fully meet PLOS ONE’s publication criteria as it currently stands. Therefore, we invite you to submit a revised version of the manuscript that addresses the points raised during the review process.

We look forward to receiving your revised manuscript.

Kind regards,

Gang Wang

Academic Editor

PLOS ONE

Journal Requirements:

Reviewers' comments:

Reviewer's Responses to Questions

**Comments to the Author**

1. Is the manuscript technically sound, and do the data support the conclusions?

Reviewer #1: Partly

Reviewer #2: Yes

2. Has the statistical analysis been performed appropriately and rigorously? 

Reviewer #1: N/A

Reviewer #2: Yes

3. Have the authors made all data underlying the findings in their manuscript fully available?

Reviewer #1: Yes

Reviewer #2: Yes

4. Is the manuscript presented in an intelligible fashion and written in standard English?

Reviewer #1: Yes

Reviewer #2: Yes

5. Review Comments to the Author

Reviewer #1: - The work reports interesting simulation results. Please include experimental results based on a real flexible spacecraft. - The over-performance of the proposed PDE-based active control needs to be quantitatively stated. Words like "superiority", "much better", "well", "is smaller" (Page 16) are very subjective and vague. How much 'better' is better? How much 'smaller' is small?- Briefly introduce a Backstepping control wth reference to literature such as 'Adaptive backstepping integral sliding mode control of a MIMO separately excited DC Motor'. Also, mention the applications of backstepping control for example, in realising fault tolerant control schemes. See the work 'Adaptive Backstepping Based Sensor and Actuator Fault Tolerant Control of a Manipulator'.- Thoroughly analyse the computational complexity of using PDE (in contrast to ODEs) in your result. This is important to show the other side of the picture, i.e., the cons of using PDEs.- What are the real-world implications of the assumptions made in this study? e.g., Assumption 1 is about the bounded nature of unknown disturbances. Is it realistic to assume in reality?- Include the version of MATLAB/Simulink used. Also, mention the specifications of the machines used to run the simulations in Section 4.- How have the parameters given in Table 1 been selected?- The paper outlines on Page 3 uses Roman numbers I, II, III, while the actual section numbering writes 1, 2, 3 and so on. Please make them consistent.- The  discussion on disturbance observer at the start of Section 3.1 could benefit from the 'Disturbance-observer-based robust control of a Serial-link robotic manipulator using SMC and PBC techniques'.-  Please thoroughly proofread the whole paper for typos, spelling mistakes and other linguistic improvements. A FEW examples are given below: - "Ploytechnic" in the affiliation of the first author -  "... which can accurately describes the ..." (describe)

Reviewer #2: The topic is valuable and the results promising, but the following improvements will strengthen the manuscript.

1. Please add a figure to clearly illustrate the specific spacecraft system or flexible structure studied.

2. The literature review should be more comprehensive. For example, similar observers for PDEs have appeared in Prof. Wenhao’s work, and Prof. Jinjun Shan has done extensive adaptive control research on flexible spacecraft. Please summarize these and other relevant works in more depth.

3. PDE modeling itself is not a novel contribution; rather, the control challenges arising from PDEs are worth deeper discussion.

4. The solution method section needs more details. MATLAB is just a platform—please further explain the actual numerical or computational techniques used.

5. The assumption on the disturbance type should be discussed.

6. PLOS authors have the option to publish the peer review history of their article (what does this mean?). If published, this will include your full peer review and any attached files.

Reviewer #1: No

Reviewer #2: No

---

## [Author Response · Author response to Decision Letter 1]

28 Jul 2025

Hello, we have made a peer-to-peer reply according to the reviewer's opinion, and the reply manuscript has been uploaded to the system. Please feel free to contact us if you have any questions.

---

## [Decision Letter · Decision Letter 1]

19 Aug 2025

Active control of flexible spacecraft in orbit based on partial differential equations

PONE-D-25-18264R1

Dear Dr. Zhang,

We’re pleased to inform you that your manuscript has been judged scientifically suitable for publication and will be formally accepted for publication once it meets all outstanding technical requirements.

Kind regards,

Gang Wang

Academic Editor

PLOS ONE

Additional Editor Comments (optional):

The reviewer is satisfied with the revision. Thus, it can be accepted.

Reviewers' comments:

Reviewer's Responses to Questions

**Comments to the Author**

1. If the authors have adequately addressed your comments raised in a previous round of review and you feel that this manuscript is now acceptable for publication, you may indicate that here to bypass the “Comments to the Author” section, enter your conflict of interest statement in the “Confidential to Editor” section, and submit your "Accept" recommendation.

Reviewer #2: All comments have been addressed

2. Is the manuscript technically sound, and do the data support the conclusions?

Reviewer #2: Yes

3. Has the statistical analysis been performed appropriately and rigorously? 

Reviewer #2: Yes

4. Have the authors made all data underlying the findings in their manuscript fully available?

Reviewer #2: Yes

5. Is the manuscript presented in an intelligible fashion and written in standard English?

Reviewer #2: Yes

6. Review Comments to the Author

Reviewer #2: It can be accepted now. But in the final version, some new papers based on PDE modeling and control should be cited, such as 10.1016/j.jfranklin.2025.107873.

7. PLOS authors have the option to publish the peer review history of their article (what does this mean?). If published, this will include your full peer review and any attached files.

Reviewer #2: No

---

## [Editor Report · Acceptance letter]

PONE-D-25-18264R1

PLOS ONE

Dear Dr. Zhang,

I'm pleased to inform you that your manuscript has been deemed suitable for publication in PLOS ONE. Congratulations! Your manuscript is now being handed over to our production team.

Kind regards,

on behalf of

Dr. Gang Wang

Academic Editor

PLOS ONE